# 46,XX DSD: Developmental, Clinical and Genetic Aspects

**DOI:** 10.3390/diagnostics11081379

**Published:** 2021-07-30

**Authors:** Camelia Alkhzouz, Simona Bucerzan, Maria Miclaus, Andreea-Manuela Mirea, Diana Miclea

**Affiliations:** 1Mother and Child Department, “Iuliu Hatieganu” University of Medicine and Pharmacy, 400012 Cluj-Napoca, Romania; calkhuzouz@umfcluj.ro (C.A.); sbucerzan@umfcluj.ro (S.B.); 2Genetic Department, Clinical Emergency Hospital for Children Cluj-Napoca, 400370 Cluj-Napoca, Romania; mary_micl@yahoo.com (M.M.); andreeamanuelamirea@gmail.com (A.-M.M.); 3Molecular Science Department, “Iuliu Hatieganu” University of Medicine and Pharmacy, 400012 Cluj-Napoca, Romania

**Keywords:** differences/disorders of sex development, 46,XX, adrenal, gonad, enzymes, genetics

## Abstract

Differences in sex development (DSD) in patients with 46,XX karyotype occur by foetal or postnatal exposure to an increased amount of androgens. These disorders are usually diagnosed at birth, in newborns with abnormal genitalia, or later, due to postnatal virilization, usually at puberty. Proper diagnosis and therapy are mostly based on the knowledge of normal development and molecular etiopathogenesis of the gonadal and adrenal structures. This review aims to describe the most relevant data that are correlated with the normal and abnormal development of adrenal and gonadal structures in direct correlation with their utility in clinical practice, mainly in patients with 46,XX karyotype. We described the prenatal development of structures together with the main molecules and pathways that are involved in sex development. The second part of the review described the physical, imaging, hormonal and genetic evaluation in a patient with a disorder of sex development, insisting more on patients with 46,XX karyotype. Further, 95% of the etiology in 46,XX patients with disorders of sex development is due to congenital adrenal hyperplasia, by enzyme deficiencies that are involved in the hormonal synthesis pathway. The other cases are explained by genetic abnormalities that are involved in the development of the genital system. The phenotypic variability is very important in 46,XX disorders of sex development and the knowledge of each sign, even the most discreet, which could reveal such disorders, mainly in the neonatal period, could influence the evolution, prognosis and life quality long term.

## 1. Introduction

Differences in sex development (DSD) in patients with 46,XX karyotype occur by prenatal or postnatal exposure to an increased amount of androgens. These disorders are usually diagnosed at birth, in newborns with abnormal genitalia, or later, due to postnatal virilization, usually at puberty. The prenatal source of excess androgens in 46,XX patients are either of foetal origin (gonadal or adrenal), either placental or maternal. The most common cause is congenital adrenal hyperplasia, which is responsible for 90–95% of these cases [1,2]. Proper diagnosis and therapy are mostly based on the knowledge of the normal development and molecular etiopathogenesis of the gonadal and adrenal structures. This review aims to describe the most relevant data that are correlated with the normal and abnormal development of adrenal and gonadal structures, in direct correlation with their utility in clinical practice, mainly in patients with 46,XX karyotype.

## 2. Steroidogenesis in 46,XX 

The main sources of steroid hormones in females are the adrenal gland, the ovary, and the peripheric tissues. One-third of female testosterone is produced in the ovary (thecal cells), the other two-thirds being synthesized in periphery tissues (by 17βhydroxysteroid dehydrogenase, type 3, and 5-17βHSD3 and 17βHSD3), starting from adrenal and ovarian precursors, mainly androstenedione (produced in equal proportions by the adrenal cortex and the ovary). In males, only 5% of testosterone is produced by the peripheral conversion of androstenedione, with 95% of testosterone being synthesized by the testes [3].

Steroid synthesis is performed under the action of steroidogenic enzymes, most of them belonging to the family of cytochromes 450, expressed in the adrenal cortex and gonads. Their effect is based on the specific transfer of electrons, in the mitochondria and endoplasmic reticulum [4]. At the mitochondria, there are type 1 enzymes, such as the following: cholesterol side channel cleavage enzyme P450 (*CYP11A1*), 11β-hydroxylase (*CYP11B1*), and aldosterone synthetase (*CYP11B2*); this is based on electron transfer using adrenodoxin reductase (Adr) and adrenodoxin (Adx) (Figure 1) [4]. At the endoplasmic reticulum, there are type 2 enzymes, such as the following: 17α-hydroxylase (*CYP17A1*), 21-hydroxylase (*CYP21A2*), and aromatase (*CYP19A1*); this is based on electron transfer using P450 oxidoreductase (*POR*) [5].

A master steroidogenic regulator is steroidogenic factor type 1 (SF1, *NR5A1*), which is involved in gene expression (most genes that are involved in steroidogenesis present at least one SF1 response element on their promotor), but also for the development of the adrenal gland and gonads. 

The quantitative regulators of acute response in steroidogenesis are the cholesterol StAR (steroidogenic acute regulatory protein) transporter system and the cholesterol side-chain cleavage enzyme (*CYP11A1*) (both in the mitochondria) [6], while long-term quantitative control is under the regulation of gene expression [4,6]. The qualitative regulator of steroidogenesis is an enzyme encoded by *CYP17A1* [4], which has both 17αhydroxylase and 17.20 lyase activity; the latter leading to C19 precursors synthesis from the C21 substrate (in the adrenal reticular area and gonads, using b5 cytochrome, expressed at the adrenarche onset) [4,6]. The B5 cytochrome mediates the interaction of POR with the enzyme encoded by *CYP17A1*, activating the lyase action and not that of 17αhydroxylase (Figure 1) [4,6]. *CYP17A1* has a higher affinity to the 17hydroxy pregnenolone substrate, and mainly influences the Δ5 pathway (conversion of dehydroepiandrosterone—DHEA—starting from 17OH pregnenolone), and less, the Δ4 pathway (conversion of androstenedione starting from 17hydroxyprogesterone) [4]. DHEA is converted to androstenedione by the 3βHSD2 enzyme (*HSD3B2*), which also catalyzes the synthesis of progesterone from pregnenolone and 17hydroxyprogesterone from 17hydroxypregnenolone. DHEAs (dehydroepiandrosterone sulfate) is the major androgen that is produced by the adrenal gland, being responsible for 95% of circulating DHEAs [7]. Androstenedione is subsequently converted to testosterone by 17βHSD3 expressed in the testis, representing the main testosterone synthesis pathway. The adrenal only expresses 17βHSD5, which allows the synthesis of smaller amounts of testosterone [4]. Classically, DHT is synthesized from testosterone mainly by 5αreductase 2 (*SRD5A2*), and expressed in the urogenital sinus, prostate primordium, genital skin, and facial and chest skin [6]. Further, 5αreductase 1 (*SRD5A1*) is expressed in non-genital skin/hair follicles, and the liver and brain; this enzyme leads to a lower level of DHT synthesis. Sixty percent of female dihydrotestosterone (DHT) is produced by the skin from androstenedione [7]. DHT has purely androgenic activity, it can no longer be transformed into other steroids. 

The non-classical pathway, which has been more recently described, explains DHT synthesis in an alternative way, starting from progesterone and 17OH progesterone, not from testosterone (Figure 1) [8,9]. It is supposed that the “backdoor pathway” mainly has a role in pathology, such as 21 hydroxylases or POR deficiency [10,11]. Estrogens are synthesized from androgens by aromatase (*CYP19A1*), expressed in the ovary, placenta, muscles, liver, hair follicle, adipose tissue, and brain [4].

Ovarian cells present different expression patterns of steroidogenic genes, thus, thecal and interstitial cells present CYP17A1 activity, but without aromatase activity (responsible for androgens synthesis), and vice versa for granulosa cells (aromatase being responsible for estrogen synthesis from androgens produced in thecal cells in the proliferative phase or progesterone in the luteal phase).

**Figure 1 diagnostics-11-01379-f001:**
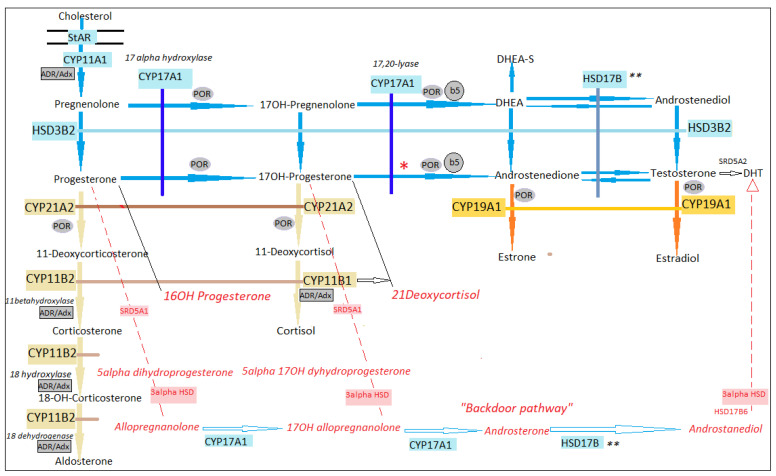
Adrenal and gonadal steroidogenic pathway. Blue area—common pathways for adrenals and gonads, cream area—observed only in the adrenal gland, orange area—observed only in the gonads, red zone—alternative pathway «backdoor pathway»; * under normal conditions 17hydroxyprogesterone is not a preferred substrate of 17αhydroxylase; ** different types of 17βHSD depending on the tissue in which it is expressed, testicle—17βHSD3, adrenal cortex—17βHSD5; abbreviation: StAR—steroidogenic acute regulatory protein; CYP11A1—cholesterol side-chain cleavage enzyme; HSD3B2—3βhydroxysteroid dehydrogenase 2; CYP17A1—17hydroxylase/17,20lyase; CYP21A2—21αhydroxylase; POR—cytochromeP450 oxydoreductase; B5—b5 cytochrome; ADR—adrenodoxin reductase; adrenodoxin—Adx; HSD17B—17βhydroxysteroid dehydrogenase; CYP11B2—aldosterone synthase; CYP11B1—11β-hydroxylase; CYP19A1—aromatase; SRD5A1—5αreductase 1; SRD5A2—5α reductase 2 [3,4,12].

The skin, by 17βHSD3 and 17βHSD3 expression, contributes to the use of DHEAs and androstenedione as precursors for testosterone synthesis in women. Adipose cells express aromatase (*CYP19A1*), which is necessary for the peripheral synthesis of estrogens from androgens.

## 3. Sex Development in 46,XX

The gonads are bipotent and undifferentiated, which is similar in the two sexes (in light microscopy, electron microscopy, and transcriptome analysis), until the age of 6 weeks of intrauterine life [13]. The gonads derive from the urogenital ridge (originated from the intermediate mesoderm in week 4), which will develop the genital, adrenal and reno-urinary structures (via pronephros, and later, mesonephros and metanephros) (Figure 2). This initial process, and the formation of the urogenital ridge, is under the influence of transcription factors (*SHH, GLI3, SALL1, FOXD2, WT1, PBX1*), signaling pathways (*WNT4*), or a telomerase activity regulator (*ACD*) [13,14]. Further development of adrenogonadal primordia is influenced by *NR5A1, NR0B1, CITED2, WNT4*, and also by vascular development [14].

### 3.1. Gonads

Gonadal primordia is observed in humans in week 5 of gestation, being under the control of *WT1, NR5A1, NR0B1, CBX1/2, LHX9, EMX2, GATA4,* and *SIX1/4* [15]. Studies in mice have shown that, at this moment, genes that are associated with Sertoli (testicular) (*SOX9, FGF9, PGD2*) or granulosa (ovarian) (*WNT4, RSPO1, CTNNB1, FST*) differentiation are expressed at similar levels in both the XY and XX fetus [16,17,18]. Depending on the presence/absence of SRY, the first gonadal cells that appear are the support cells, Sertoli cells in 46,XY, or granulosa cells in 46,XX. These cells will lead to a gender-specific differentiation, further inducing the differentiation of steroidogenic cells, Leydig cells in the 46,XY fetus, respectively, thecal cells in the 46,XX fetus [13,15,17,19,20].

In 46,XX fetuses, the gonads remain undifferentiated for a longer time compared to 46,XY, due to a later expression of the ovarian differentiation genes (*WNT4, RSPO1, CTNNB1, FST*) [15,17]. Although Alfred Jost’s initial theory of a passive pathway to ovarian differentiation has long been embraced, active mechanisms of ovarian differentiation have been shown [21]. Thus, in humans, unlike rodents, the absence of SRY does not lead to ovarian differentiation, but gonadal dysgenesis, without a meiotic progression of oocytes, thus indicating the need for active ovarian differentiation associated with further active stabilization [3,15].

The differentiation of germ cells depends on the cellular environment (ovarian or testicular). In females, primordial germ cells differentiate into oogonia in week 8. In week 10, the primordial follicles develop, the oocytes being surrounded by a single layer of granular cells (these cells influencing further maturation of the oocyte), and the germ cells enter into the first meiosis. In week 15, the primary follicles develop, thus thecal cells are observed. In weeks 23–24, the first de Graaf follicle is observed [3]. These processes develop variably for each germ cell; however, by the end of the seventh month of gestation, most germ cells have entered meiosis I, but a lot of them will degenerate through atresia until term, and the viable cells will remain until ovulation, which is initiated after gonadal maturation at puberty [3]. Germ cells are critical for maintaining ovarian stabilization; in their absence, ovarian follicles will degenerate and lose the ability to synthesize hormones. Thus, in females, the gonads will acquire a streak appearance, unlike the males, for whom the loss of germ cells does not affect somatic and Leydig cells [22,23].

*WNT4* has an anti-testicular effect, directly, through the inhibitory action on SOX family proteins, and indirectly, by stabilizing beta catenin, which accumulates in the nucleus and interacts with *LEF1* to inhibit *SOX9* and to activate *FST* [13,15] (Figure 3). *FST* also has an anti-testicular action, by inhibiting activin B, which is the important molecule in early testicular differentiation, by influencing testicular vascularization. *RSPO1* both stimulates *WNT4* expression and stabilizes beta-catenin, which are considered to be key molecules in ovarian determinism or suppression of testicular formation [24]. At the same time, *WNT4* has a pro-ovary effect by stabilizing germ cell survival.

*FOXL2* is a gene that is expressed in the developing eyelid mesenchyme, as well as in the ovary (fetal and adult granulosa cells), being an important marker in early ovarian differentiation, especially in stimulating follicular development [15]. This gene is also expressed in gonadotropic and thyrotropic cells. In prenatal and postnatal mice, *FOXL2* deletion stops follicular maturation before primary follicle development, and induces subsequent atresia and also a male differentiation of the ovaries, by stimulating SOX9 expression (by reactivating TESCO), thus transforming support cells into Sertoli-like cells [3,15]. *FOXL2* also inhibits SF1 expression by antagonizing *WT1* during ovarian development in mice, and stimulating aromatase in granulosa cells [3,15]. *FOXL2* suppresses testicular gonadal development in females and is considered the equivalent of *DMRT1* in males (known as a suppressor of ovarian development, and with a role in maintaining testicular differentiation) [25,26].

*MAP3K1* is involved in the balance between female and male development, having a role in stabilizing beta-catenin by sequestering *AXIN1* (a gene that is upregulated by *SOX9* and *FGF9*), which normally blocks ovarian development by promoting beta-catenary destabilization [3,15,27].

*NR0B1* (coding for DAX1) is the first known testicular repressor potential, suggested by the duplication of this gene and sex reversal in 46,XY [13]. *NR0B1* deletion does not influence ovarian differentiation in 46,XX. This gene is also involved in the development of the adrenal cortex, and mutations with a loss of function are associated with congenital adrenal hypoplasia [28].

**Figure 3 diagnostics-11-01379-f003:**
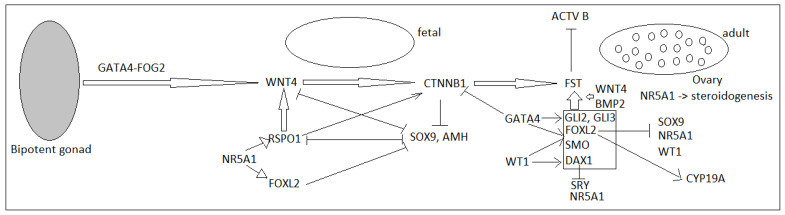
Molecular factors involved in ovary development [29].

### 3.2. Internal Genital Organs

The internal genital tract develops from two pairs of ducts, derived from the intermediate mesoderm, the Wolff ducts (mesonephric duct), and the Muller ducts (paramesonephric) (lateral and parallel to Wolff); starting in week 5, these ducts are developed from cranial to caudal. At the lower end of the Wolff duct, the Muller duct crosses it, becoming medial, joining the contralateral Muller duct at this level to later form the uterus [3,13].

The internal genitals are also similar for both sexes until week 9, and will differentiate into male or female tractus depending on gonadal determinism and differentiation. Thus, testicular differentiation leads to the synthesis of testosterone (Leydig) (from weeks 9–10) and anti-Mullerian hormone (AMH) (Sertoli) (from week 12), which leads to the stabilization of the Wolff duct to form the epididymis, vas deferens, and seminal vesicles (under the action of androgens) and, respectively, to the craniocaudal regression of the Mullerian ducts [3]. In females, there is no need for ovarian hormone synthesis to regulate the differentiation of the internal genitalia. Due to the absence of AMH, the Muller ducts differentiate into the oviduct, uterus, cervix, and upper vagina, and the absence of androgens leads to the craniocaudal regression of the Wolff duct until complete regression [3].

### 3.3. External Genital Organs

The external genitalia is developed from the urogenital sinus (formed from the cloaca in week 7), constituted by the urogenital membrane (disappears in week 9), surrounded laterally by genital folds (derived from cloaca folds, represented by the labioscrotal and urethral folds) and anteriorly by the genital tubercle [3,30,31]. The external genitalia is similar for both sexes until week 9 of gestation.

Virilization of the external genitalia normally occurs in 46,XY fetuses, starting in week 9, due to exposal to testicular androgens synthesis (locally metabolized to DHT), with scrotum formation by hypertrophy and posterior fusion of the labioscrotal folds, and penile gland development from the genital tubercle (in weeks 9–12), and subsequent growth of the penile gland throughout pregnancy [3,31,32]. Female external genitalia development occurs without hormonal stimulation, being developed from week 11 until week 20. The genital tubercle differentiates towards the clitoris, the urogenital sinus remains open by the disappearance of the urogenital membrane, with the opening of the anterior urethra and posterior vagina, within the vestibular portion of the sinus. The vestibule is bordered laterally superficially by the urethral folds, which become the labia minora, and in depth by the bulbs of the vestibule, the labioscrotal folds remain unfused, posteriorly forming the major labia.

At 10 weeks, in both XX and XY fetuses, there is a vaginal primordium formed by the fusion of the caudal prominence of the fused Muller ducts and the sinovaginal bulb (at the posterior region of the urogenital sinus), forming the vaginal plaque, which subsequently permeabilizes, forming the vaginal lumen in the lower two thirds [3]. The differentiated genital tubercle in the female sex, towards the clitoris, will lead to a structure made up of the clitoral glans, clitoral corpus cavernosum (the equivalent in males), and two clitoral cavernous pillars, as well as the vestibular bulbs (the equivalent of spongy bodies in males) (Figure 4). In 46,XX fetuses, virilization only occurs pathologically by exposure to DHT, depending on the amount and timing of exposure during pregnancy [3,30].

External genitalia evolution is influenced by exposure to androgens in a critical period, called the masculinization programming window, which is common to both sexes, and is considered to be between 8 and 14 weeks of gestation [30]. It seems that WNT signaling pathways have a role in determining this window. In humans, a good marker of virilization of the external genitalia is the anogenital distance between the anus and the base of the genital gland, which is about twice as large in men as in women, due to the action of fetal androgens (Figure 5) [30]. Anogenital distance is a non-invasive postnatal marker of androgen exposure in the masculinization windows in males, which is important as a predictive marker for hypospadias, cryptorchidism, oligospermia, or testicular cancer [33,34,35]. In women, androgen exposure in this window can lead to prostate and seminal vesicles development, large anogenital distance, and clitoris hypertrophy, but with the impossibility of stabilizing and maintaining the Wolff duct (maybe this process requires higher androgen exposure compared to the urogenital sinus), while later exposure only influences the size of the genital tubercle [30]. Early modelling of the external genitalia is under the control of factors that regulate the interactions between mesenchymal (genital fold) and epithelial tissues (covering urogenital sinus structures), with *SHH* and beta-catenin being essential in the formation of bipotential genital primordia, and *BMP*, *FGF* and *HOX* gene families having an important role in genital tubercle development [36].

## 4. Adrenal Development

The adrenal gland originates from the intermediate mesoderm (cortical) and neuroectoderm (medullary). The adrenal–genital primordium later gives birth to the adrenal primordia, from which the fetal adrenal cortex and the gonadal primordia will be formed; the separation of the two structures takes place in week 7. The adrenal structure is differentiated due to increased expression of *NR5A1, NR0B1, WNT4*, and *CITED2* [14]. In week 7, the fetal adrenal gland is made up of two structures, cortical and medullary, with the medullary being formed from the migrated sympathoadrenal cells (cells of the neuroectoderm). At the age of 8 weeks, the adrenocortical primordia is differentiated into two distinct layers, the fetal zone, internal (FZ), and the definitive, external zone (DZ) (differentiated until week 12), and they completely encapsulate the medullary region at the age of 9 weeks (Figure 5) [38,39]. The definitive area later differentiates into the adult adrenal gland. During the second trimester, the fetal area greatly increases in volume, and is responsible for increasing the fetal secretion of DHEA and DHEAs, then this fetal area progressively regresses, leading to decreased secretion of these hormones [7]. After birth, the definitive area develops in the fasciculate area (internal) and glomerular area (external), and after the age of 2 years, the most internal area proliferates, giving birth to the reticular area, which is initially constituted by small islands, which synthesized smaller amounts of DHEA and DHEAs around the age of 3 years. The reticular area develops progressively into a thickened layer, which is responsible for the significant increase in the secretion of DHEA and DHEAs at the adrenarche moment, around 7–8 years (Figure 6) [7,38]. This differentiation in layers is made according to the expression of various transcription factors (SF1, Pref1/ZOG, inner zone antigen), with the expression of various hormones in these layers depending on the enzyme expression involved in their synthesis.

In adulthood, the glomerular area is distributed over about 15% of the cortex, containing small cells with small nuclei arranged in nests [7]. The fasciculate area is the largest, comprising about 75% of the cortex, with large cells, loaded with lipids (clear cells), arranged radially, in cords and the rest glomerular area, with irregular cells, with small amounts of lipids, but rich in granules of lipofuscin [7]. The progenitor cell population is located between the glomerular and fasciculate area, its migration and cell differentiation occur in the fasciculate area, and senescence occurs in the reticular area. Fetal cells give rise to subcapsular stem cells that differentiate into deep areas (centripetal) [14].

The glomerular area is regulated by the renin-angiotensin system and potassium levels. At this level, aldosterone is synthesized, due to the expression of the *CYP11B2* gene (aldosterone synthetase). Aldosterone synthetase has 11 beta hydroxylation, 18 hydroxylation, and 18 oxidation activity, thus allowing the conversion of 11 deoxycorticosterone to corticosterone, 18-hydroxycorticosterone and, subsequently, to aldosterone [4,6]. The enzyme 17 alpha-hydroxylase is not expressed, thus not allowing the synthesis of cortisol and androgens.

The fasciculate and reticulated areas are regulated by ACTH; here, cortisol, androgens and small amounts of estrogen are synthesized. Unlike the glomerular area, CYP11B2 is not expressed here, making aldosterone synthesis impossible. An ACTH deficiency leads to atrophy of the fasciculate and reticulated area, and an excess of ACTH leads to the hypertrophy of this region. In general, the fasciculate area responds acutely to the increase in ACTH, and the reticular area is responsible for the basal, chronic secretion, regulated by ACTH [7]. A prolonged stimulation leads to the depletion of lipids in the fasciculated area and, over time, to an aspect of the reticulated area (aspect of the reticulated area that extends to the capsule), and also to a transformation of the glomerular area to a fasciculate structure [4].

The reticulated area does not allow the synthesis of the Δ4 pathway; a small amount of androstenedione is observed, but a massive amount of DHEAs, as the 3βHSD2 expression is small, but of DHEA sulfotransferase is high. With adrenarche onset, the reticulated area begins to secrete large amounts of DHEA, DHEAs, and androstenedione, but with low testosterone production (17βHSD3 is unexpressed in the adrenal cortex, only 17βHSD5 is expressed here, leading to a small quantity of testosterone synthesis) [7].

### Differences in Sex Development in 46,XX

DSD are classified, depending on karyotype, into the following three categories: 46,XX DSD, 46,XY DSD, and sex chromosomes abnormalities DSD [40,41].

The 46,XX DSD are classified as follows: 1. disorders of gonadal differentiation (testicular DSD—SRY positive, SOX9 duplication; ovotesticular DSD; primary ovarian insufficiency due to genes involved in gonadal development—FSH receptor mutation, NR5A1, WT1; syndromic forms); 2. disorders with excessive amounts of androgens, such as fetal cause—congenital adrenal hyperplasia—21 hydroxylase deficiency, 11 beta hydroxylase deficiency, 3 beta hydroxysteroid dehydrogenase deficiency; virilizing ovarian tumors; virilizing adrenal tumors; glucocorticoid receptor gene mutation, POR; placental cause—aromatase deficiency, POR; maternal cause—luteoma, exogenous; 3. Others, such as Mayer–Rokitansky–Küster–Hauser (MRKH) syndrome—type I and II; complex syndromic disorders; cloacal exstrophy; Mullerian duct agenesis; vaginal atresia; labial fusion [40,41].

## 5. Clinical Assessment

The incidence of genital abnormalities is about 1 in 5000 newborns [40]. Clinical features that draw attention to a sexual development abnormality in the newborn are the following: isolated clitoral hypertrophy, isolated posterior hypospadias, bilateral cryptorchidism or ectopia, unilateral cryptorchidism/testicular ectopia added hypospadias or micropenis [40,42]. At puberty, clinically suggestive signs for DSD may be indicated by virilization of the external genitalia, pubertal delay, or primary amenorrhea [43].

Clinical assessment includes a precise description of the size of the genital tubercle, presence or absence of labioscrotal folds fusion, the number and localization of orifices, and the presence or not of palpable gonads at labioscrotal folds. Based on these data, the Prader scale is used to assess the degree of sexual ambiguity [41,42] (Figure 7), as follows: stage I—clitoromegaly without labial fusion; stage II—clitoromegaly and posterior labial fusion, without urogenital sinus; stage III—important clitoromegaly (penoclitoral organ), almost complete fusion of the labial folds a single urogenital orifice (urogenital sinus) with perineal opening; stage IV—penile organ, complete labial fusion, urogenital sinus with an opening at the base or on the ventral surface of the penile gland; stage V—penile organ, scrotum appearance (similar to the male sex, without palpable gonads), urethral meatus at the top of the penile gland [44].

The external masculinization score can also be calculated, by giving a score to each of the following: glans size (score from 0—micropenia, depending on the standard deviation score, to 3—normal penis size), labioscrotal fusion (score from 0—without fusion to 3—with fusion), gonads position (a score is given for each testis, 0—abdominal or absent on examination, 1—inguinal and 1.5—labioscrotal) and location of the urinary meatus (0—proximal, 1—mid, 2—distal and 3—normal) [43].

The measurement of anogenital distance is another important marker, used to assess androgen exposure during intrauterine life (Figure 5) [30,37]. In each patient, the measurement of arterial tension is necessary, and is very useful in cases with congenital adrenal hyperplasia.

## 6. Hormonal Assessment

First-line examinations in a newborn with ambiguous external genitalia primarily targets congenital life-threatening adrenal cortical hyperplasia by determining 17-hydroxyprogesterone (not relevant values in the first 36 h), serum electrolytes (usually without changes in the first 4 days of life), and glycemia [43]. First-tier tests also include anatomical and genetic evaluations (karyotype + *SRY*). If there is a strong suspicion of congenital adrenal hyperplasia, therapy can be initiated, not before preserving blood and urine for further serum analysis of other steroids (testosterone, DHT, androstenedione, plasma renin activity, deoxycorticosterone, DHEAs, and others) and urinary/24-h evaluation of urinary steroid profile, which can give clues for the diagnosis of different adrenal enzymatic blocks [43]. On the other hand, the mini puberty could be added to large fluctuation of gonadotrophins and androgenic steroids in the first months of life, leading to the need for subsequent evaluation after mini puberty.

In the case of a patient with 46, XX karyotype and without palpable organs, a higher value of 17-hydroxyprogesterone is suggestive for congenital adrenal hyperplasia. Usually, a value above 10 mg/l for 17 hydroxy progesterone indicates an enzymatic block, most commonly given by 21 hydroxylase deficiency, and a value below 2 mg/L excludes this diagnosis, with the intermediate values recommending a stimulation with synthetic ACTH (an increase over 10 mg/l confirming the diagnosis) [46].

A measurement of 17-hydroxyprogesterone is recommended to be performed before 8 o’clock, and after the onset of menstrual cycles; this assessment should be conducted in the follicular phase [46]. Also, 21-deoxycorticosterol is a useful marker that is associated with 17-hydroxyprogesterone, to diagnose 21-hydroxylase deficiency. Glucocorticoid deficiency is indicated by higher ACTH, with hyperpigmentation of the genitals and nipples, and also by hypoglycemia or growth retardation. Plasma renin activity is useful in determining aldosterone deficiency in congenital adrenal hyperplasia, and will distinguish between the salt-losing form and simple virilization. The urinary steroid profile may be useful in confirming a diagnosis of 21-hydroxylase deficiency or in identifying other possible enzymatic deficiencies (Table 1).

The analysis of plasma/serum steroids is indicated to be performed by immunoassay or liquid chromatography associated with tandem mass spectrometry (LC-MS/MS) [4]. The analysis of urinary steroids is conducted by gas chromatography associated with mass spectrometry (GC/MS), a method with high specificity, evaluating even the unusual steroid metabolites that are obtained in rare enzymatic deficits (Table 1) [4,47].

If patients with karyotype 46,XX have palpable gonads, a priori diagnosis is ovotesticular DSD and, less frequently, testicular DSD, with the quantity of testicular tissue being correlated with AMH and even testosterone levels. Gonadotrophins are also useful; however, the final diagnosis is given by the histopathological examination.

## 7. Anatomic Assessment

A clinical evaluation of the external genitalia is always followed by an imaging evaluation, to describe the characteristics of the urogenital sinus, Mullerian structures, gonads, adrenal glands, and sometimes renourinary structures (due to the common embrionary origin) [43,48]. The initial evaluation consists of an ultrasound examination. It does not always allow easy visualization of dysgenetic or fibrous gonads or testicles with intra-abdominal location, as well as the prepubertal uterus in pubertal children (sometimes requiring reassessment after estrogen therapy) [43].

When ultrasound evaluation fails to detail the internal structures, particularly Mullerian derivatives or dysgenetic/ectopic gonads (e.g., ectopic testes are better visualized if they are in extra-abdominal localization) or urinary tract abnormalities, MRI is indicated [48,49]. It targets the pelvis and perineum, and sometimes the abdomen (to visualize adrenal gland or tumors).

Genitography can also be useful in characterizing the internal genital ducts, but is often replaced by genitoscopy (endoscopic evaluation of the genital tract), to better characterize the urogenital sinus, Mullerian structures, and their relationship to the urethra [43].

Laparoscopy is especially indicated when the gonads should be more specifically assessed (visualization, sampling of biopsy fragments, and gonadectomy of intra-abdominal structures) [50], but it is not so efficient for a fine observation in the profound pelvis, such as the identification of discrete Mullerian derivatives that are closely attached to the bladder [43].

MRI evaluation remains an election investigation to define and detail the anatomy of the gonads and internal genitalia, with laparoscopic evaluation remaining a last option [49]. It is always necessary to consult a pediatric surgeon, who will better indicate an anatomical evaluation.

## 8. Genetic Assessment

Genetic evaluation in DSD is firstly based on the result of chromosome analysis and the SRY gene (Figure 8). Thus, in the case of 46,XX DSD with SRY negative, level 17 of hydroxyprogesterone (basal or after stimulation with synthetic ACTH) will be the next in the diagnostic evaluation algorithm. An elevated level argues in favor of congenital adrenal hyperplasia, and the first etiology is the deficiency of 21-hydroxylase, so the first intention is to evaluate this gene by classical sequencing Sanger and MLPA, as it is difficult to evaluate by next-generation sequencing, due to the presence of a very similar pseudogene.

If no changes in CYP21A2 are observed, or if the 17-hydroxyprogesterone value is not altered, other genes will be evaluated, coding for other enzymes that are less commonly associated with congenital adrenal hyperplasia or involved in (ovo) testicular 46,XX DSD, by genes panel or exome/genome sequencing. If, for a patient 46,XX DSD, congenital adrenal hyperplasia was excluded, the molecular studies mainly target the genes SRY, SOX9, SOX3, SOX10, RSPO1, or WNT4.

Sometimes the genetic evaluation of the peripheral blood is not sufficient in interpretation, and it is necessary to assess the genetic and histological characteristics of the gonadal tissue, to establish the etiological diagnosis, but also the tumor risk that is associated with gonadal dysgenesis. Additionally, sometimes it might be necessary to evaluate gene expression and gonadal regulation patterns, by determining mRNA (by quantitative PCR techniques or RNASeq), by assessing methylation status (by specific PCR or array technique), or by the evaluation of chromatin changes (by chromatin immunoprecipitation techniques—ChIP).

The association of other clinical features to DSD indicates a syndromic form of the pathology, which is usually due to a greater change at the genomic level, and thus indicates the evaluation of copy number variants (CNVs), either by chromosomal analysis by microarray (SNP array or CGH array) or by bioinformatics analysis of large structural variants using sequencing data [52].

## 9. 21 Hydroxylase Deficiency

### 9.1. Frequency

Congenital adrenal hyperplasia is the most common cause of 46,XX DSD, with 21-hydroxylase deficiency being observed in 95% of cases. It is present in 1:15.000 newborns, with a higher incidence in some isolated populations, such as Yupiks Eskimos in Alaska, affecting 1:300–700 newborns [53].

### 9.2. Etiopathogenesis

The 21-hydroxylase deficiency is caused by *CYP21A2* gene mutations (6p21.3). This gene has a pseudogene in its proximity, *CYP21A1P*, with about 98% homology, thus favoring recombination between both genes, and therefore the occurrence of deletions/duplications with detrimental effects (20% of patients) [54]. Large structural variants usually induce a more severe disease phenotype. Single-nucleotide variants (SNVs) are also an important cause of this disease. Depending on the residual level of the enzyme, the clinical severity may be variable. This gene has an autosomal recessive inheritance, but cases of heterozygous patients with attenuated phenotypes have been described, similar to non-classical forms [55]. Deficiency of this enzyme induces a metabolic block of aldosterone and cortisol synthesis, with the impossibility to convert progesterone to deoxycorticosterone and 17-hydroxy-progesterone to 11-deoxycortisol, with the substrates being derived to the synthesis of the androgens (Table 1). Deficiency in cortisol synthesis leads to ACTH stimulation.

### 9.3. Clinical Picture

Three types are described, two classic types, the simple virilizing and the “salt-wasting” type, with neonatal diagnosis and one non-classical, with an attenuated phenotype, usually diagnosed at puberty.

A cardinal feature of the classic forms in patients 46,XX is the variable virilization of the external genitalia, due to hyperandrogenism. In all these 46,XX cases, the gonads are not palpable, which must draw attention when establishing social sex in a newborn with masculinized genitalia.

About 75% of classic cases have a severe enzymatic deficiency of 21-hydroxylase, represented by the “salt-wasting” type, which leads to a significant deficiency of aldosterone in the first weeks of life (4–15 days after birth), which is clinically validated by renal salt loss, vomiting, dehydration with hyponatremia and hyperpotassemia, metabolic acidosis, and potentially fatal hypovolemic shock. In all the classical forms, there is aldosterone deficiency, indicated by increased plasma renin activity, but the clinical phenotype of mineralocorticoid deficiency is only evident in forms of severe enzyme deficiency, in the “salt-wasting” type.

In the simple virilizing form, the enzymatic deficiency is partial, yet inducing a cortisol deficiency, which stimulates ACTH synthesis that is responsible for adrenal hyperplasia and amelioration of cortisol levels, but with an increase in 17-hydroxyprogesterone shifted to androgen synthesis, causing external genitalia virilization. In the absence of therapy, virilization continues in both sexes, causing pubic pubarche, hirsutism, even in infants, accelerated skeletal maturation with premature closure of the growth cartilage (“tall children and short adults”), muscular hypertrophy, and lower voice tone (Figure 9).

The increase in 17-hydroxyprogesterone in healthy individuals has low metabolic efficiency, but when it is elevated, it has an anti-mineralocorticoid effect, leading to an increase in plasma angiotensin and renin, which can restore sodium balance.

The non-classical type, with late onset, is due to mild enzymatic deficit, which does not lead to virilization during embryo–fetal development, with the clinical picture being obvious only at puberty.

Hyperandrogenism leads to virilization of variable intensity (early puberty, mild clitoral hypertrophy, hirsutism, acne), accelerated rate of growth and bone maturation, and menstrual disorders (primary or secondary amenorrhea or oligomenorrhea/bradimenorrhea) [56]. The non-classical forms discreetly affect cortisol synthesis, and generally do not lead to Addisonian crisis. Recent studies suggest a frequency of a non-classical form of 1:200 in the general population [57]. The diagnosis is based on hormonal (Table 1), anatomic and genetic assessment [4]. Further, 17-hydroxyprogesterone and 21-deoxycortisol are useful markers for diagnosis [58].

### 9.4. Treatment

The goals of treatment are to supplement the cortisol deficiency, and thus to interrupt the feedback that stimulates the gland, and to ameliorate hyperandrogenism [46]. Hydrocortisone is preferred in the treatment of children, and the dose is 15–20 mg/m^2^/day divided into three sub-doses, respecting the circadian rhythm of cortisol secretion (higher morning dose, respectively, lower evening, e.g., 10–7.5–5 mg/day), with a 2–3 times dose increase in the cases of stress, trauma, surgery, or acute illness. Substitution treatment with mineralocorticoids is usually made with fludrocortisone, and the dose is 0.15–0.3 mg/m^2^/day (0.05–0.2 mg/day). the results of modified-release hydrocortisone (MR-HC) preparations are very promising, especially in the case of two doses, at 22 h and 8 h, with a better cortisol control overnight than hydrocortisone in three doses [59]. Antiandrogens, such as cyproterone acetate or spironolactone, 50–100 mg/day orally, may also be added. Excess hydroxyprogesterone, in the case of untreated 21-hydroxylase deficiency, has an anti-mineralocorticoid effect; this effect is more obvious in the case of severe and improperly treated patients [60]. In severe virilization of 46,XX, the surgical treatment that is recommended is vaginoplasty using urogenital mobilization, and for severe clitoromegaly a neurovascular-sparing clitoroplasty [1]. In minimally virilized girls at birth, early surgical treatment may be proposed. However, a delayed surgery and observation until an older age should also be considered [1]. An early intervention could be favorable for anatomic resolution, a better development of gender identity, and better future reproductive health. However, it also has some disadvantages, due to functional or sexual sequels risk or the possible need for recurrent interventions. The early surgical treatment makes it impossible to respect a patient’s decision [1,61]. Thus, an extensive discussion with the parents is always needed, about the advantages and risks of early or delayed surgical intervention [1,61]. Surgical treatment of minors should be performed with full informed consent of the parents, and, in the case of an older children, also with their assent, after detailed discussions regarding the risks and benefits and potential complications.

Important data about the psychological aspects in DSD are provided by a large study, dsd-LIFE, on 1040 patients with DSD, including 221 women with 46,XX congenital adrenal hyperplasia [62]. Due to this large sample, this study supports the evidence-based recommendations for the improvement of therapy and care of DSD patients. The study observed that most 46,XX DSD patients with congenital adrenal hyperplasia have quite a good quality of life, which could be greatly influenced by the individual’s health status, and the care should mostly be oriented around physical or psychological chronical problems [63]. Additionally, it was seen that the number of gender changes in DSD patients is increased, but for not very high (5% of the population studied) [64]. However, a problem could be their sexuality, the majority are not satisfied with their sex life, having a large spectrum of sexual problems, with adult patients often needing special care in this direction [65].

### 9.5. Prenatal Diagnostic and Treatment

Prenatal diagnosis can be made by CYP21A2 genetic testing of fetal biologic samples (chorionic villosities or amniocentesis). Circulant fetal DNA analysis is also very useful to diagnose fetal sex, and to exclude the male fetus from prenatal treatment, this test could be done precociously (after 6 weeks of gestation, with better results at 9–10 weeks). In pregnant women who are at risk of having a fetus with congenital adrenal hyperplasia, a low dose of dexamethasone 20 μg/kgc/day (or 0.5–1.5 mg/day) could be administered, but it is recommended to regard prenatal therapy as experimental, as the risk and benefits are not very well defined due to the lower number of patients from whom the outcome is well reported [1]. Treatment is continued until a villous chorion biopsy (week 10–12), circulatheADN analysis (week 10), or amniocentesis (week 16–18) can be performed, then, depending on fetal sex and disease diagnosis, the decision to continue or not with the therapy will be taken.

### 9.6. Neonatal Screening

In many countthe, the programs of newborns screening existents decreased the rate of complications in 21-hydroxylase deficiency, particularly in the forms with “salt wasting”, thus lowering the risk of infant mortality and morbidity [1]. The screening is based on the 17-hydroxyprogesterone measurement (as a first-tier test), and the second-tier test is represented by LC-MS/MS [1]. On the other hand, the newborns screening for 21-hydroxylase deficiency is not performed in each country, thus delaying the diagnosis in a number of patients, with a higher implication on morbidity and mortality, and even on an appropriate choice of social sex.

## 10. 11 Beta Hydroxylase Deficiency

### 10.1. Frequency

The 11β-hydroxylase deficiency is the second leading cause of congenital adrenal hypertrophy, and is observed in approximately 5% of these patients (1:100,000 newborns) [66,67].

### 10.2. Etiopathogenesis

The enzyme 11β-hydroxylase (encoded by *CYP11B1,* expressed in the fasciculate area) is involved in the metabolism of deoxycortisol to cortisol. This enzyme deficiency is responsible for cortisol deficiency, and thus increased ACTH levels, without mineralocorticoid deficiency. Thus, the hypertrophy of the gland occurs, and the deviation of the unmetabolized substrate in excess towards the androgenic metabolism leads to the virilization of girls. The enzyme deficiency causes the accumulation of 11-deoxycortisol (compound S) (limited biological activity) and 11-deoxycorticosterone (with mineralocorticoid activity) [4].

### 10.3. Clinical Picture

Prenatal hyperandrogenism, in the absence of treatment, induces the continuation of virilization in both sexes, and thus early isosexual pseudopuberty in boys and heterosexual in girls, with infancy onset and represented by pubarche development, hirsutism, android muscle hypertrophy, decreased tone of voice, and accelerated growth with a short final height (Figure 10). High blood pressure is observed and, in some cases, the values may exceed 200 mmHg [67,68].

It was also observed that a non-classical form (with late onset) is validated only in girls, at puberty.

The diagnosis is based on hormonal (Table 1), anatomic and genetic evaluation [4].

### 10.4. Treatment

Except for mineralocorticoid substitution, which is not required for 11β-hydroxylase deficiency, the treatment and monitoring of this pathology are similar to that of patients with 21-hydroxylase deficiency. Hypokalemic hypertension with low renin may sometimes be encountered, and then calcium channel blockers (captopril) are recommended.

## 11. 3-βHSD Type 2 Deficiency

### 11.1. Etiopathogenesis

There are two isoenzymes of 3-βHSD, type 1 and type 2, which differ by 23 amino acids. Type 1 is expressed in the liver, skin, placenta or prostate, and type 2 is expressed exclusively in the adrenal and gonads. Further, 3-βHSD type 2 deficiency is found in less than 0.5% of patients with congenital adrenal hyperplasia, and is due to impaired metabolization of Δ5 steroids into Δ4 steroids, thus influencing all three corticosteroid hormonal lines with a decreased synthesis of mineralocorticoids, glucocorticoids and androgens [69]. The Δ5 steroids (17-hydroxypregnenolone, DHEA, DHEAs) have low androgenic activity, but in the case of excess accumulation will lead to virilization of the external genitalia in the 46,XX patient.

### 11.2. Clinical Picture

In 46,XX, at birth clitoral hypertrophy is often observed and sometimes labial posterior fusion, and in the older children acne, hirsutism, premature puberty, advanced bone growth and early closure of growth cartilage, and in adult women, hirsutism, a certain degree of clitoromegaly, and sometimes polycystic ovaries are observed. It is known that Δ5 steroids in excess can be metabolized in the periphery by 3-βHSD type 1, thus explaining a paradoxically higher level of 17-hydroxyprogesterone. This enzyme block can also affect estrogen synthesis in girls, and thus there is the need for estrogen replacement to induce and maintain secondary sexual characteristics and menstrual cycles [69]. Decreased mineralocorticoid synthesis leads to varying degrees of salt loss, sometimes requiring a substitute treatment. Newborns with a complete loss of function mutation may have adrenal insufficiency in the neonatal period. A non-classical form was also observed, with late onset.

The main steroid marker that is used to establish the diagnosis, is noticed in Table 1 [2,4]. In the context of adrenal and gonadal deficits, there is an increase in ACTH, FSH, and LH.

Often these patients require hormone replacement of the three lines.

## 12. POR Deficiency

### 12.1. Etiopathogenesis

The protein that is encoded by POR acts as an electron donor from NADPH to microsomal steroidogenic enzymes, and influences the synthesis of glucocorticoids and sex hormones by influencing the action of the enzymes encoded by *CYP21A2, CYP17A1,* or *CYP19A1* [70]. Skeletal abnormalities could also be observed, due to the deficient POR interaction with enzymes that are involved in sterol synthesis (CYP51A1, SQLE, CYP26A1, CYP26B1, CYP26C1). POR deficiency thus leads to impaired steroidogenesis (cortisol and sex hormones—leading to DSD) and skeletal malformations with a similar phenotype to Antley-Bixler syndrome. The impairment of glucocorticoid synthesis is generally partial, an assessment of basal cortisol being normal, but with an abnormal response to stress, and there may also be an increase in the level of mineralocorticoids that are responsible for the possible high blood pressure in these patients [71].

### 12.2. Clinical Picture

In 46,XX, POR deficiency is observed as DSD or large ovarian cysts, and sometimes a pubertal delay. Virilization in females is explained by the metabolization of excessive 17-hydroxyprogesterone to “the backdoor pathway” (Figure 1), with the increase in DHT and other androgens [4]. A peculiarity of this pathology compared to other forms of congenital adrenal hyperplasia, is the lack of progressive postnatal virilization. If the fetus is affected, maternal virilization could be observed during pregnancy, manifested by acne, hirsutism or clitoral hypertrophy, due to excess androgens that are produced by the “backdoor pathway”. The skeletal changes are similar to those seen in Antley-Bixler syndrome (pathology due to the FGFR2 mutation), represented by the following: craniosynostosis, middle face hypoplasia, proptosis, choanal atresia, low-set ears with auditory canals stenosis, synostosis of large joints (radiohumeral), bowing of long bones, neonatal fractures, joint contractures arachnodactyly, and congenital clubfoot [70].

Laboratory traits in POR deficiency are observed in Table 1, and the hormonal profile is sometimes consistent with a deficiency of 21-hydroxylase or 17alpha-hydroxylase/17.20 lyase [4].

Prenatal diagnosis can be established by evaluating the mother’s urinary steroids [72].

Treatment involves glucocorticoid replacement and sex hormone therapy (estrogen in females).

## 13. Glucocorticoid Receptor Deficiency

It is caused by mutations with a loss of function in the glucocorticoid receptor (NR3C1), leading to glucocorticoid resistance (increased cortisol, but no clinical signs of hyperfunction) and increased ACTH levels, which leads to the stimulation of adrenal cortical hormone synthesis (aldosterone, cortisol and androgens) and the clinical picture represented by hypertension, hypokalemia, female virilization, premature pubarche, and hirsutism [3]. Treatment is based on the appropriate administration of synthetic glucocorticoids.

## 14. Maternal Androgens Excess

In general, the 46,XX fetus is protected from excess maternal androgens by placental aromatization into estrogen; however, sometimes a degree of virilization can be observed if the mother was exposed, during pregnancy, to androgens or progestogens of exogenous origin (e.g., norethindrone, etisterone, noretinodrel, medroxyprogesterone acetate, or danazol) [2,3]. Other sources of maternal hyperandrogenism could be an ovarian tumor (hilar cell tumors, arrhenoblastoma, lipoid cell tumor, Krukenberg tumors) or the adrenal tumor (less often, but possible during pregnancy) [2,3]. In the case of congenital adrenal hyperplasia of the mother, placental aromatization prevents virilization of the 46,XX fetus. In some situations, endocrine disruptors must be considered, knowing that they could influence different hormonal pathways.

## 15. Pregnancy Luteoma

It is a benign tumor of the ovary, with a low incidence, appearing during pregnancy, often in the second trimester. It occurs through the marked proliferation of luteinized cells, under the action of bHCG, leading to increased synthesis of progesterone and androgens; the latter being responsible for the virilization of the 46,XX fetus and the mother [73]. It is often an incidental discovery during an ultrasound examination, and can sometimes be complicated by bleeding, ovarian torsion, and mass effect. It is a tumor that can cause major problems of differential diagnosis with a malignant tumor. Usually, it suffers a spontaneous postpartum regression. Testosterone and dihydrotestosterone dosing is useful for diagnosis.

## 16. Aromatase Deficiency

### 16.1. Etiopathogenese

The fetal adrenal gland produces significant amounts of 17-hydroxypregnenolone and 16-hydroxy DHA, which are further converted in the placenta to androgens and estrogens. Aromatase produces the conversion of androgens C19 to estrogen C18, with important roles in the placenta and postnatally, as a key enzyme in the synthesis of estrogen. Placental aromatase deficiency leads to increased levels of androgens that return to the fetal circulation and lead to the virilization of 46,XX patients [3].

### 16.2. Clinical Picture

At birth, 46,XX patients have varying degrees of virilization, and later they will develop pubertal delay, absence of telarche, polycystic ovaries, hypergonadotropic hypogonadism, amenorrhea, and decreased bone mineral density [3]. Treatment is based on estrogen replacement. Pregnancy with an affected fetus also leads to maternal hyperandrogenism and hypoestrogenemia, which are changes that are responsible for maternal progressive virilization.

## 17. 46,XX DSD by Gonadal Differentiation Abnormalities

### 17.1. Testicular DSD

It is characterized by the presence of testes in 46,XX patients (but with azoospermia and subsequent testosterone deficiency), absent Mullerian derivatives, and normal or sometimes ambiguous external genitalia (15% of cases) [3,74]. The prevalence of this pathology is 1:20,000, and 90% of these patients present the SRY gene. A less common cause is the presence of chromosomal rearrangements or large structural variants involving SOX9, SOX3 or SOX10 genes (protesticular genes), usually duplications leading to overexpression [3]. Social sex is almost always male. These patients will need testosterone replacement therapy. Infertility is often the reason why these patients are evaluated in adulthood. In children, testicular hypoplasia and short stature can be observed at puberty (testicular volume is associated with the volume of Sertoli cells, which is normal until puberty, but further associated with lower testicular volume due to azoospermia) [74].

### 17.2. Ovotesticular DSD

It is defined by the following three situations: (1) testicular tissue (seminiferous tubules) and ovarian tissue (mandatory follicular structures containing oocytes) in each of the two gonads (bilateral ovotestis); or (2) one testis on one side and ovary on the other; or (3) one ovotestis on one side and ovary or testis on the other. Usually, the testicular tissue is dysgenetic, and the ovarian tissue is normal. This disorder is often associated with chromosomal changes, such as mosaics 46,XX/46,XY, in other situations with 46,XX karyotype, and extremely rare in cases with 46,XY [3].

The clinical phenotype depends on the percentage of ovarian and testicular tissue. Thus, if the predominance is ovarian, the phenotype is of a feminized newborn, but with clitoral hypertrophy and possible posterior fusion of the labial folds. If the preponderance is testicular, the newborn is rather male, but with possible signs of hypovirilization (hypospadias or cryptorchidism). In ovotesticular 46,XX DSD, contrary to testicular 46,XX, 90% of patients are SRY negative. However, similar to testicular DSD, there is overexpression of protesticular genes (*SOX9, SOX10, SOX3*), or deficit of those pro-ovarian genes (*RSPO1, WNT4*) [3]. *NR5A1* and *WT1* mutations were also described in association with ovotesticular or testicular 46,XX [75,76]. RSPO1 mutations are associated with 46,XX sex reversal, testicular, or ovotesticular DSD, the absence of Mullerian derivatives, and associate palmoplantar hyperkeratosis and squamous cell carcinoma. Heterozygous mutations of *WNT4* in 46,XX are responsible for a milder phenotype, and are associated with hyperandrogenism (stimulates the steroidogenic enzyme expression, including SRD5A2), abnormal development of Mullerian derivatives, but with normal external genitalia, and sometimes with primary amenorrhea. Homozygous mutations are responsible for 46,XX DSD (sex reversal XX, with testicular or ovotesticular DSD) adrenal, renal and pulmonary dysgenesis (SERKAL syndrome), which is a syndrome with lethality in intrauterine life [3,15].

### 17.3. 46,XX Gonadal Dysgenesis

46,XX gonadal dysgenesis is a primary ovarian defect, either due to a developmental abnormality or to resistance to gonadotropin stimulation, and leads to premature ovarian failure.

Mutation of the FSH receptor is a very rare cause of gonadal dysgenesis in 46,XX patients, leading to hypergonadotropic ovarian failure, through premature depletion of the follicular reserve, streak gonads, pubertal delay, primary amenorrhea, and infertility [77]. Low bone mineral density is observed due to estrogen deficiency. Other genes that are associated with gonadal dysgenesis in 46,XX patients are as follows: *NR5A1, BMP15, PSMC3IP, MCM9, SOHLH1, NUP107, MRPS22*, and *ESR2* [78,79].

## 18. Conclusions

Most patients with 46,XX DSD present as etiology congenital adrenal hyperplasia, by enzyme deficiencies that are involved in the hormonal synthesis pathway. The other cases are explained by gene abnormalities that are involved in the development of the genital system. The phenotype variability is very important in 46,XX disorders of sex development, and the knowledge of each sign, even the most discreet, which could reveal such disorders, mainly in the neonatal period, could influence the evolution, prognosis and life quality long term. More conclusive data are observed in collaborative clinically oriented projects, such as dsd-LIFE, especially on more debated aspects, such as psychosocial adaption and well-being, psychosexual development, improvement in clinical care, patients and parents view, ethics, and cultural context.

## Figures and Tables

**Figure 2 diagnostics-11-01379-f002:**
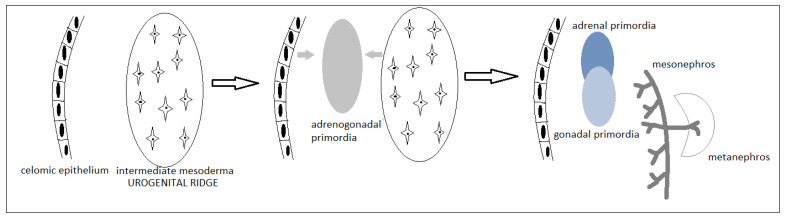
Development of gonadal, adrenal and renourinary primordia in week 4 (first two stages in figure)—week 5 (third stage) [14].

**Figure 4 diagnostics-11-01379-f004:**
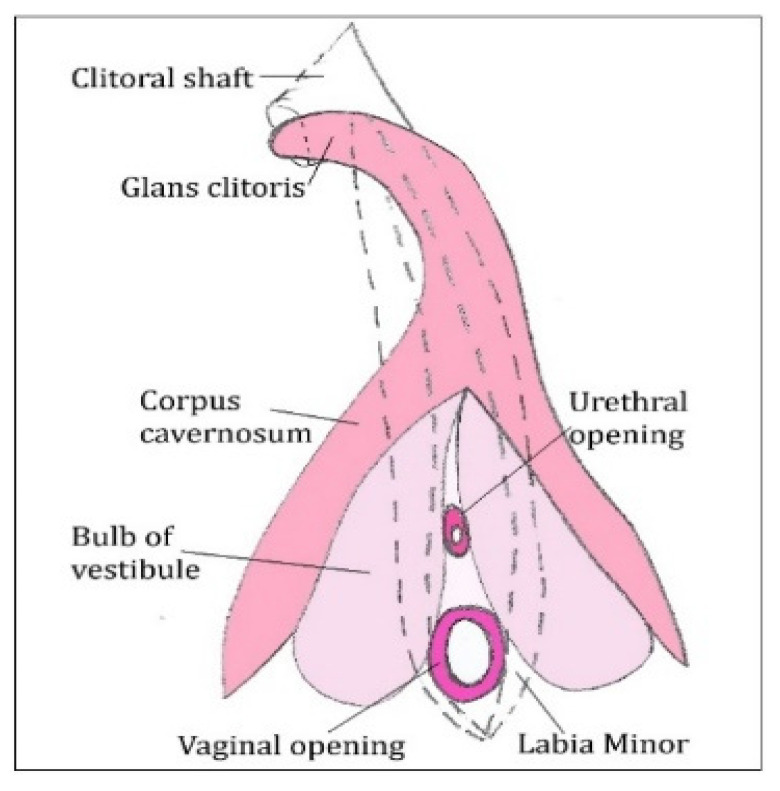
Clitoris morphology.

**Figure 5 diagnostics-11-01379-f005:**
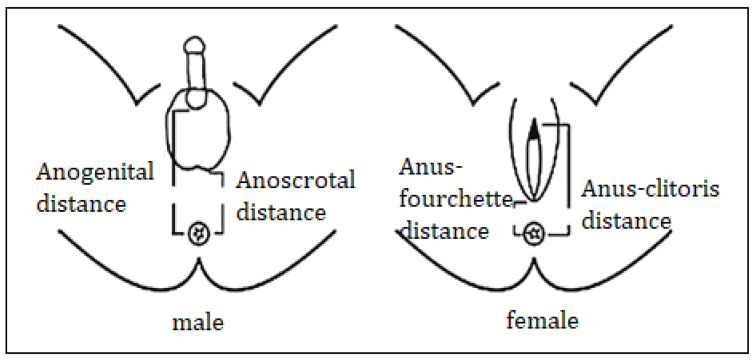
Measurement of different anogenital distances, markers of fetal androgens exposure [37].

**Figure 6 diagnostics-11-01379-f006:**
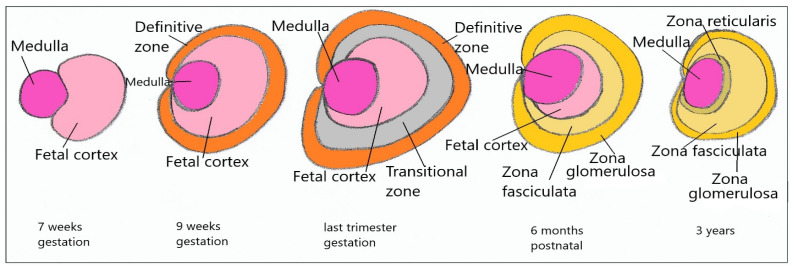
Adrenal gland development.

**Figure 7 diagnostics-11-01379-f007:**
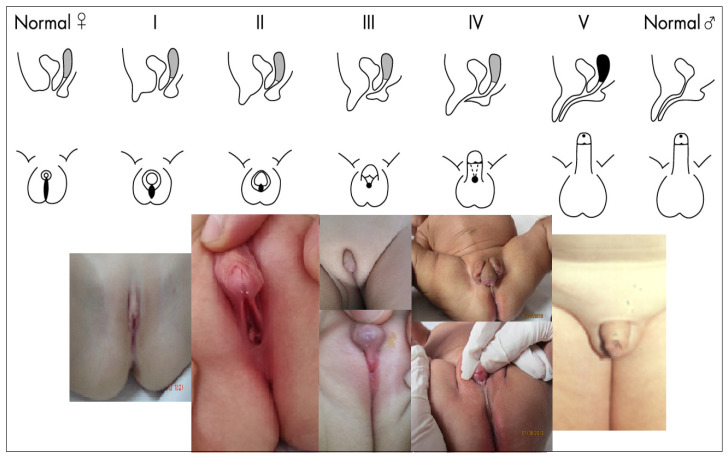
Prader stages with clinical examples for each stage [45]. Written informed consent was obtained from the parents for publication of this photos.

**Figure 8 diagnostics-11-01379-f008:**
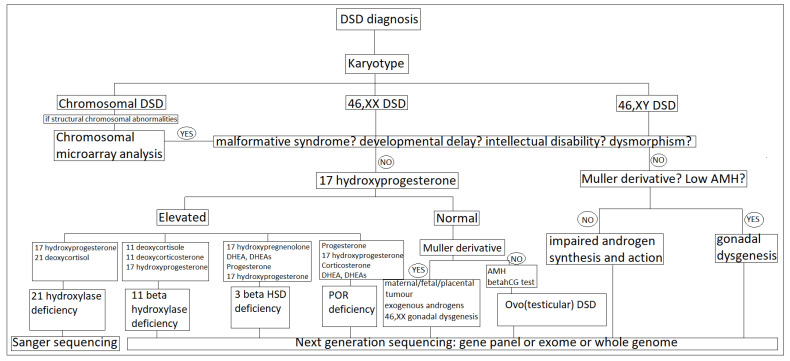
Genetic testing algorithm in DSD [51].

**Figure 9 diagnostics-11-01379-f009:**
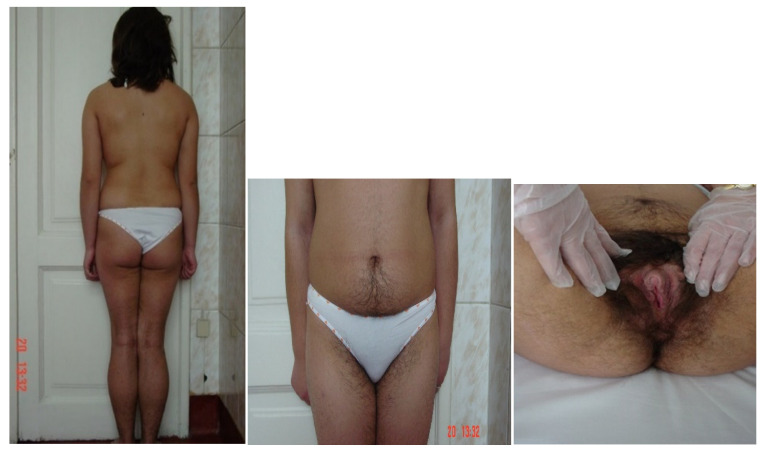
46,XX DSD patient with simple virilizing form of 21-hydroxylase deficiency (Prader 2). Written informed consent was obtained from the parents for publication of these photos.

**Figure 10 diagnostics-11-01379-f010:**
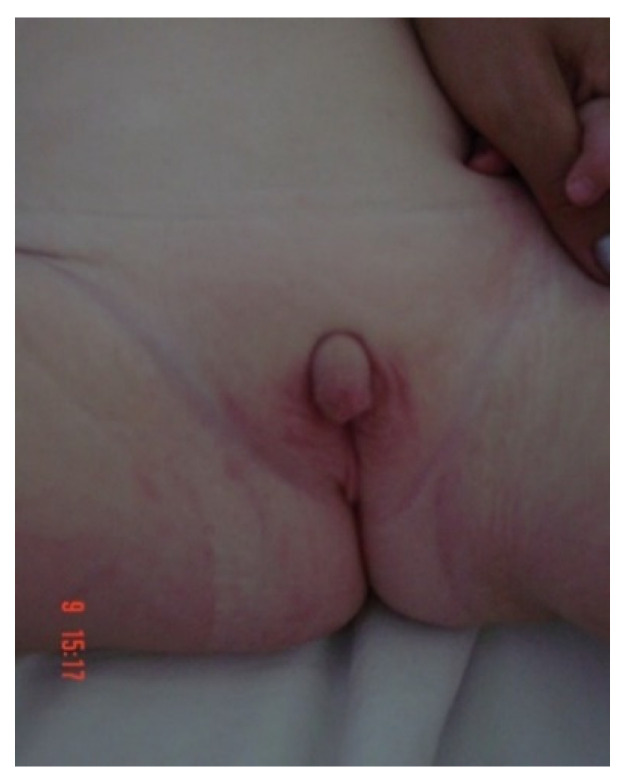
46,XX DSD in a patient 6 months of age with 11β-hydroxylase deficiency. Written informed consent was obtained from the parents for publication of this photo.

**Table 1 diagnostics-11-01379-t001:** Serum/plasma and urinary steroids in different disorders of 46,XX DSD [4].

Disorder	Etiology	Serum/Plasma Steroids	Urinary Steroids
21-hydroxylase deficiency	CYP21A2	↑17-hydroxyprogesterone ↑21-desoxycorticosterol ↑DHEAs, ↑androstenedione, ↑testosterone ↓aldosterone, ↑ plasma renin activity, ↓deoxycorticosterone ↓11-deoxycortisol, ↓corticosterone, ↓cortisol	↑pregnanetriol (17-hydroxyprogesterone metabolite) ↑pregnanetriolone (21-desoxycorticosterol metabolite) ↑pregnanetriolone/tetrahydrocompoundF (cortisol metabolite) ↑17-ketosteroid/24 h
11β-hydroxylase deficiency	CYP11B1	↑11-Deoxycorticosterol ↑11-Deoxycorticosterone ↓plasma renin activity ↑androsteondione ↑testosterone N/↑17-hydroxyprogesterone	↑tetrahydrocompoundS (11-deoxycortisol metabolite) ↓tetrahydrocompoundF ↑tetrahydrocompoundS/tetrahydrocompoundF ↑17-ketosteroid/24 h
3-βHSD deficiency	HSD3B2	↑Δ5-steroids-17-hydroxypregnenolone,DHEA, DHEAs ↑17-hydroxypregnenolone/17-hydroxyprogesterone ↑DHEA/androstendione ↑17-hydroxyprogesterone (synthesised from 17-hydroxypregnenolone via HSD3B1)	↑5-pregnenetriol (17-hydroxypregnenolone metabolite) ↑androstenetriol (DHEA metabolite) ↑5-pregnenetriol/tetrahydrocompound F ↑androstenetriol/tetrahydrocompound F ↑pregnanetriol, 17α-hydroxypregnanolone (17-hydroxyprogesterone metabolites)
POR deficiency	POR	↑progesterone ↑17hydroxyprogesterone ↑21desoxycorticosterol ↑corticosterone, ↑deoxycorticosterone ↓DHEA ↓DHEAs	↑pregnanetriol, 17α-hydroxypregnanolone ↑pregnanetriolone ↑pregnanediol (pregnenolone, progesterone, deoxycorticosterone metabolites) ↓androsterone, ↓etiocholanolone (DHEA, androstendione, testosterone metabolites)
Aromatase deficiency	CYP19A1	↓estriol, estrone, estradiol ↑androstendione,testosterone,DHT	-
Ovotesticular or testicular 46,XX DSD	SRY, SOX9, SOX3, SOX10 NR5A1, RSPO1, WNT4	↑AMH ↑testosterone	-

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
