# Peer review of "46,XX DSD: Developmental, Clinical and Genetic Aspects"

_diagnostics, 2021, doi:10.3390/diagnostics11081379_

Round 1

Reviewer 1 Report

Please see the attached review.

Author Response

Dear Reviewer,

Thank you for revising our article, it is very useful for a better paper. Further, I would try to answer to the suggestions and questions you made.

  1. In my opinion the Authors should choose what is the main aim of the manuscript. At the moment it seems partially like a book chapter. I think the Authors could concentrate on the new, current knowledge in DSD field and then the paper would be more interesting and valuable.

I modified the structure of the paper and information, following the main aim: to expose the main actual data from etiopathogeny to treatment in 46,XX,DSD, in order to better help in diagnostic and treatment of these disorders

  1. Title does not give the main information about the paper. We have both the basic information about gonadal and adrenal development, not only in patients with 46,XX karyotype.

I modified the paper to be more concentrated on 46,XX.

  1. Paragraphs regarding gonadal and adrenal development should be more concise with underlying the new information in the field.

I modified this part to be more concise and actual.

  1. I cannot see the references regarding the very important European dsd-Life programme.

I add this reference

  1. The paragraphs regarding the CAH diagnosis and treatment should include new methods: the possibility of neonatal screening for CAH in some countries (line 406); clinical trials regarding new long lasting steroid forms (lines 591-595). The dexamethasone treatment during pregnancy should be commented as not recommended by all endocrine centres (lines 604-609).

I modified this paragraphs

  1. Line 420: the recommendation for karyotype examination should be underlined (before discussing different karyotype results)

I modified, according to your suggestion

  1. Line 241: age 3 (years?) is not the age of adrenarche start. If we understand adrenarche as first clinical characteristics it is 8 yrs in girls and 9 yrs in boys. We do know that the adrenal glands initiate increased adrenal production earlier, at the age of 6-8 yrs, some authors consider the earlier age. It should be clearly commented.

I modified to be more clearly, it was a translation error.

  1. The English language needs major corrections in grammar and style. Some parts of the

manuscript are difficult to understand. Please find my suggestions:

- line 18: “imagistic” please change to “imaging”

- line 20: delete “are”

- line 43: “pronephrosis”?

- lines 45-47, 54-57, 60, 99; Fig 2, 6, 8: explain abb. (please check the whole manuscript)

- lines 85-86, 139-140, 237, 238-243, 351-353: unclear

- line 354: I would recommend changing “are produced” to “result from”

- line 368: 2 x ”syndrome”

- line 384: “scala” to “scale”

- line 420: “second intention” to “secondary”

- line 598: “associated” to “added”

- line 757: “by abnormalities” to “result from affected”

- line 516, 610, 643, 757, 799: there shouldn't be a dot (“.”) in enzyme deficit classification or karyotype results

I modified and verified all these.

  1. Overall, in my opinion the manuscript should be more concise and the Authors should underline the new relevant information in the DSD field more clearly.

I change the manuscript according to your suggestions.

Reviewer 2 Report

The review article by Camelia et.al. on differences of sex development in patients with 46XX is an important study and the authors have discussed in detail the physical, genetic and hormonal evaluation of disorders of sex development, stressing more on patients with 46XX karyotype. While this is an important study, significant amount of language editing is required and the manuscript cannot be accepted in the present form. There are many long sentences which should be shortened for easy understanding and interpretation. For e.g. line 69-72 is too long and feels like several lines from different articles have been merged into one single sentence. Also, it is difficult to interpret what authors are trying to say in line 113-116 or line 188-192, and I request to re-frame these sentences, or break down to two or more sentences. The conclusion is confusing too and I request to change the language. Few other needed edits: -Line 34: Change "very much" to "mostly" -Line 74: Remove "will" in the sentence " will lose the ability to" -Line 102-112: Shorten the paragraph -Line 117: Remove contribute or in involved, cannot be both "MAP3K1 contributes is involved..." -Line 121-125: reduce repeating NR0B1 in every line and replace with either "The gene" or "It" -Line 236-238: Please re-frame the sentence and avoid using periods followed by question mark "this foetal area progressively involves postnatal...?, leading to decreased" -Line 811: Never start a sentence with a numerical digit, either spell it out or re-frame the sentence -Line 813: Change " genes abnormalities" to "genetic abnormalities" -Line 814: Change "phenotype variability" to "phenotypic variability"

Author Response

Dear Reviewer,

Thank you for revising our article, it is very useful for a better paper. Further, I would try to answer to the suggestions and questions you made.

While this is an important study, significant amount of language editing is required and the manuscript cannot be accepted in the present form. There are many long sentences which should be shortened for easy understanding and interpretation. For e.g. line 69-72 is too long and feels like several lines from different articles have been merged into one single sentence. Also, it is difficult to interpret what authors are trying to say in line 113-116 or line 188-192, and I request to re-frame these sentences, or break down to two or more sentences. The conclusion is confusing too and I request to change the language.

I modified according to your suggestions. I checked the English language with the help of an editing service.

Few other needed edits: -Line 34: Change "very much" to "mostly"

I modified, according to your suggestion.

Line 74: Remove "will" in the sentence " will lose the ability to"

I modified, according to your suggestion.

Line 102-112: Shorten the paragraph

I modified, according to your suggestion.

Line 117: Remove contribute or in involved, cannot be both "MAP3K1 contributes is involved..."

I modified, according to your suggestion.

Line 121-125: reduce repeating NR0B1 in every line and replace with either "The gene" or "It"

I modified, according to your suggestion.

Line 236-238: Please re-frame the sentence and avoid using periods followed by question mark "this foetal area progressively involves postnatal...?, leading to decreased"

I modified, according to your suggestion.

Line 811: Never start a sentence with a numerical digit, either spell it out or re-frame the sentence

I modified, according to your suggestion.

Line 813: Change " genes abnormalities" to "genetic abnormalities"

I modified, according to your suggestion.

Line 814: Change "phenotype variability" to "phenotypic variability"

I modified, according to your suggestion.

Round 2

Reviewer 1 Report

Please see the attached review.

Author Response

Dear Reviewer,

Thank you again for this revision. In the following, I will answer to your observations.

  • “in te face of abnormal genitalia” in Abstract, Introduction- please modify, it is not the best expression.

I modified.

  • “In minimally virilised girls at birth surgical treatment could be performed precociously a delayed surgery and observation until the child is older should be considered [1]. A precocious intervention could be favourable for anatomic resolution, a better development of gender identity, better future reproductive health, however due to functional or sexual sequels risk or recurrent interventions needed in some situation or the need to choose an intervention depending on patient decision, the intervention could be delayed [1][61].”- unclear, please modify.

I modified.

  • “Neonatal screening. Due to neonatal screening by dosing 17 hydroxyprogesterone (as a first tier test) in many countries, these complications are less obvious and thus decrease the risk of infant mortality and morbidity in forms with “salt wasting”[1]. As a second-tier test is LC-MS/MS [1]. However newborn testing for 21 hydroxylase deficiency is not performed in each country and the diagnosis could be sometimes delayed, with higher implication even on informed choosing of social sex.”- unclear, please modify.

I modified.

  • “More conclusive data are expected to be observed in collaborative clinically oriented projects, such as dsd-LIFE especially on more debated aspects such as psychosocial adaption and well-being, psychosexual development, improvement of clinical care, patients and parents view, ethics and cultural context [76].”- please add this information to Discussion, not for the first time in Conclusions. Please comment in more detail on the programme.

I modified. I added more data in text.

Kind regards,

Reviewer 2 Report

This is a very important study and authors have made sufficient changes as previously suggested. The manuscript can be accepted in the present form.

Author Response

Dear Reviewer,

Thank you a lot for your revision.

Kind regards,

Round 3

Reviewer 1 Report

Please see the attached review.

Author Response

Dear Reviewer,

Thank you again for your kind suggestions.

“in the face of abnormal genitalia” has been changed in Abstract, but not in Introduction (first paragraph)- please modify. As the naming in DSD is quite challenging we need to be careful, the word “ambiguous” is not recommended any more. I would suggest just writing (Abstract, Introduction) “These disorders abnormalities are usually diagnosed at birth in newborns with ambiguous or abnormal genitalia or later, due to postnatal virilization, usually at puberty.”

I modified according to your suggestion.

Please clarify further the paragraph regarding the surgical intervention possibilities. We cannot write about “precocious surgical treatment”, I would suggest to modify the text as follows:
“In minimally virilised girls at birth female newborns early surgical treatment could may be proposed to be performed precociously but il. However, a delayed surgery and observation until the older age it also should be also considered a delayed surgery and observation until the child is older should be considered [1]. A precocious An early intervention could be favourable for anatomic resolution, a better development of gender identity, better future reproductive health. However, but it also have some disadvantages, due to functional or sexual sequels risk or , the possible need for recurrent interventions needed in some situation. The early surgical treatment makes it impossible or the need impossibility an intervention depending on to respect a patient’s decision., the intervention could be about his body [1][61]. Thus an extensive discussion with the parents is always needed, about advantages and risks of early precocious or delayed surgical intervention [1][61].

I modified according to your suggestion.

Thank you!

Kind regards